# Water-Insoluble, Thermostable, Crosslinked Gelatin Matrix for Soft Tissue Implant Development

**DOI:** 10.3390/ijms25084336

**Published:** 2024-04-14

**Authors:** Viktória Varga, László Smeller, Róbert Várdai, Bence Kocsis, Ibolya Zsoldos, Sara Cruciani, Renzo Pala, István Hornyák

**Affiliations:** 1Institute of Translational Medicine, Semmelweis University, 1094 Budapest, Hungary; varga.viktoria.dora@semmelweis.hu; 2Department of Materials Science and Technology, University of Győr, 9026 Győr, Hungary; kocsis.bence@sze.hu (B.K.); zsoldos@sze.hu (I.Z.); 3Department of Biophysics and Radiation Biology, Semmelweis University, 1094 Budapest, Hungary; smeller.laszlo@med.semmelweis-univ.hu; 4Laboratory of Plastics and Rubber Technology, Department of Physical Chemistry and Materials Science, Faculty of Chemical Technology and Biotechnology, Budapest University of Technology and Economics, 1111 Budapest, Hungary; vardai.robert@vbk.bme.hu; 5Research Centre for Natural Sciences, Institute of Materials and Environmental Chemistry, 1111 Budapest, Hungary; 6Department of Biomedical Sciences, University of Sassari, Viale San Pietro 43/B, 07100 Sassari, Italy; scruciani@uniss.it (S.C.); renzopala6@gmail.com (R.P.)

**Keywords:** crosslinked gelatin, biomaterials, scaffold development, material science

## Abstract

In this present study, the material science background of crosslinked gelatin (GEL) was investigated. The aim was to assess the optimal reaction parameters for the production of a water-insoluble crosslinked gelatin matrix suitable for heat sterilization. Matrices were subjected to enzymatic degradation assessments, and their ability to withstand heat sterilization was evaluated. The impact of different crosslinkers on matrix properties was analyzed. It was found that matrices crosslinked with butanediol diglycidyl ether (BDDE) and poly(ethylene glycol) diglycidyl ether (PEGDE) were resistant to enzymatic degradation and heat sterilization. Additionally, at 1 *v*/*v* % crosslinker concentration, the crosslinked weight was lower than the starting weight, suggesting simultaneous degradation and crosslinking. The crosslinked weight and swelling ratio were optimal in the case of the matrices that were crosslinked with 3% and 5% *v*/*v* BDDE and PEGDE. FTIR analysis confirmed crosslinking, and the reduction of free primary amino groups indicated effective crosslinking even at a 1% *v*/*v* crosslinker concentration. Moreover, stress–strain and compression characteristics of the 5% *v*/*v* BDDE crosslinked matrix were comparable to native gelatin. Based on material science measurements, the crosslinked matrices may be promising candidates for scaffold development, including properties such as resistance to enzymatic degradation and heat sterilization.

## 1. Introduction

The use of scaffolds is one of the main aspects of tissue engineering and regenerative medicine (TERM). Generally, the intended use of scaffolds is to provide a viable environment for the growth of cells and tissues [1]. The appearance of the scaffolds can be liquid, gel-like, or solid, but in all cases, it is required to be compatible with the extracellular matrix (ECM) and has to be suitable to be used in a three-dimensional application [2,3]. The use of cells and growth factors in TERM is also a popular approach; however, in our case, we aimed to enable the scaffold to eventually become a commercialized medical device. Thus, to keep the regulatory requirements in mind, we generally avoided the addition of cells and growth factors. The fabricated matrices in TERM are generally applied directly into or onto the defect, and the regenerative processes are allowed to take place according to the natural healing steps [4].

In order to enable a newly developed scaffold to be regulated as a medical device, we have to fulfill the applicable quality-management-related standards [5]. One of these requirements is to use materials that have already been investigated and found to be safe for human implantation [6]. This means that the materials are known to be biocompatible, and the degradation of these materials is well-known and considered to be safe. Another important requirement for the application as a soft tissue implant is to have a pore size distribution that is suitable for cells to grow; the material must have a relatively high surface, be non-toxic, be biodegradable, and have mechanical properties that closely resemble soft tissue. The degradation profile is also an important characteristic. Our aim was to develop a solid matrix that can be implanted for a longer period of time; thus, our material must not dissolve in water and must not melt under physiological conditions, especially pH and temperature [7]. Besides the mechanical properties, the scaffolds need to support the viability of cells. Preferably, the cells can proliferate and migrate in the inner structure to allow for differentiation and remodeling in the long term. These materials can be produced using a variety of techniques, including phase separation, rapid prototyping, leaching, electrospinning, freeze drying, and centrifugal casting. In our case, freeze drying was chosen because the solvent is water and high temperature is not required; thus, biomimetic materials are optimal for this preparation method [8]. The contents of the scaffolds are generally polymers, which can be synthetic or natural. The most important aspects from our viewpoint are biocompatibility and biodegradability. Furthermore, the scaffold needs to preserve its mechanical integrity and promote cell attachment, viability, and natural regenerative functions as part of the ECM and soft tissue milieu [9]. In order to have a material that closely resembles soft tissue, we chose a material that is derived from collagen. Collagen is the most abundant protein in the human body, with nearly 1/3 of the whole protein content; hence, the starting material for our experiments was a partially degraded collagen: gelatin [10]. Gelatin is a biodegradable polymer that has a protein content of 85–92%. This material is non-toxic, does not induce immunological reactions, and is generally suitable for cell adhesion and viability [11,12]. Gelatin, due to its versatile applicability, has been used for over a millennium in the food, cosmetic, and pharmaceutical industries. Due to the partial hydrolysis of collagen, gelatin is a thermoreversible material that is soluble in water at a physiological temperature and can form solid gels at approximately 35 °C. However, this can depend on the original collagen source. Gelatin from mammals has a higher helix–coil transition temperature, but, for example, that from cod is below 15 °C [13]. This low melting point is the main disadvantage of utilizing gelatin as a scaffold, as it would melt instead of exhibiting slow and steady degradation [10]. In order to improve mechanical properties, gelatin can be blended to form composites [14], and/or can be further modified with the modification of the functional groups or with crosslinking [15]. The crosslinkers used include glutaraldehyde, genipin, formaldehyde, and 1-(3-dimethylaminopropyl)-3-ethyl-carbodiimide hydrochloride. Furthermore, the crosslinking may also be performed with the use of UV crosslinkers [16] or enzymatically [17]. The functional groups that can be utilized for crosslinking are generally hydroxyl, carboxyl, amino, or thiol groups [18]. In our case, these functional groups are all present in the monomers of gelatin. Therefore, it would be useful to apply a crosslinker that has been used for over a decade and is generally considered safe, as are the degradation products. In our previous works, we also found that the use of butanediol diglycidyl ether (BDDE) is safe, as the biocompatibility and cell viability did not change with the use of this material [19]. BDDE was also used to crosslink fish-derived gelatin, and the mechanical properties were examined but not as a biologically applicable material [20]. It had also been discovered earlier that BDDE is suitable to crosslink primary amino groups, and it has been used to modify collagen from sheep [21]. However, in this case, both the starting material and the end product were water-insoluble. Our aim was to crosslink gelatin to prepare a water-insoluble, heat-stable material and to investigate the characteristics of this material. Additionally, another family of epoxy crosslinkers contains poly(ethylene glycol) building blocks, which is called poly(ethylene glycol) diglycidyl ether (PEGDE). This material was also tested as a potential crosslinker. Vinyl sulfone derivatives are also known as effective crosslinkers in biomaterial design [22]. In our case, divinyl sulfone (DVS) was also used, which has been effective and was also effective in our previous experiments when hyaluronic acid was the starting material [19]. The scope of the present investigation was to use the novel approach of applying solid freeze-dried gelatin pads that are placed in a crosslinker matrix, and the process of optimizing the reaction parameters and the evaluation of the crosslinked matrices.

## 2. Results

### 2.1. Degradation of Native Gelatin in Solutions with Different pH Values

Two-way ANOVA with Turkey’s post hoc test was performed, and the different groups were compared to each other after 168 h. The degradation of native gelatin was lowest in PBS, which was significantly lower compared to all the other groups. The fastest degradation was observed in the case of TRIS8 and TRIS9 solutions. The degradation in these buffers was significantly higher compared to all other groups. However, there was no significant difference between the degradation in the two TRIS buffer solutions. Surprisingly, there was no significant difference between the degradation in water compared to NaOH12 after 168 h (Figure 1).

The experiment was conducted to find out the stability of GEL and to find the optimal solvent or buffer type for the crosslinking and to compare it to the ones found in the scientific literature. The goal of the optimization process was to find the optimal pH for the crosslinking, which is in the alkaline range, and to avoid the further degradation of GEL. According to Figure 1, the optimal pH was 12, with the use of NaOH, and the optimal reaction time was 48 h, as after 24 h the crosslinked weight was lower than the starting weight. However, in terms of scaffold development, sterility is mandatory, and we planned to use the most convenient method: heat sterilization. Unfortunately, the crosslinked product that we produced in the pH = 12 NaOH solution further degraded during heat sterilization; thus, we moved to a 1% NaOH solution, which we found to be optimal during our previous work [19]. The addition of 1% NaOH solution to the GEL samples under the same circumstances led to the total degradation of the matrices in 4 h. Surprisingly, the simultaneous addition of 1% NaOH and BDDE or PEGDE to GEL resulted in a stable, water-insoluble matrix after 48 h, which was able to withstand heat sterilization without losing integrity or shape. When DVS was used, we found that the crosslinked matrices were not able to withstand heat sterilization either, and the crosslinked matrix partially degraded. Thus, we moved on with testing the 1% NaOH solution with BDDE and PEGDE for the use of the production of crosslinked GEL.

### 2.2. Weight Differences and Swelling Ratio

Generally, the used crosslinkers can be harmful for the human body; thus, it is important not to leave any unreacted crosslinkers in the scaffold and to use as little crosslinker amount as possible while maintaining the required stability. In a preliminary experiment, we found that the swelling ratio did not further decrease above a 5 *v*/*v* % crosslinker. Thus, we measured the starting and the crosslinked weights, as well as the swelling ratio to find out the optimal crosslinker reagent amount (Figure 2).

According to the results, the crosslinked weight of the scaffolds was higher than the starting weight when 3% and 5 *v*/*v* % was added to the scaffold. Furthermore, the swelling ratio became lower as the amount of crosslinker increased. However, the crosslinked weight was only significantly higher when 3% and 5 *v*/*v* % PEGDE was used compared to 1 *v*/*v* % PEGDE; the swelling ratios did not differ significantly from each other in either group.

### 2.3. Enzymatic Degradation Differences

Based on the weight differences, the stability of the native 5% GEL in water and in collagenase solution was compared to the crosslinked 5% GEL using 20 *v*/*v* % BDDE (Figure 3). The expectation was that the crosslinked samples would degrade less than the native ones, and that those in water would degrade only slowly.

The aqueous samples (GH) showed relatively little degradation, with low absorbance even after 48 h. However, there was a significant difference between the native gelatin samples in water and in collagenase (GC) solution after 2 h. From 4 h, there was a significant difference between the GH and GC groups and between the GC and crosslinked GEL in the collagenase solution (XGC) group. This difference remained significant even after 48 h. Thus, with this method, we demonstrated that the crosslinking was successful and that BDDE is capable of stabilizing gelatin properly. However, the crosslinking took place with the use of 20 *v*/*v* % BDDE, so we planned to further investigate the optimal crosslinking parameters. The enzymatic degradation was tested with the use of reduced BDDE and with 0.2 mg/mL collagenase (Figure 4).

As it is visible in the figure, according to our expectations, the highest degradation was in the 1 *v*/*v* % PEGDE-containing matrix (50G1P). It was followed by the 1% BDDE-containing one (50G1B), and then the 3 *v*/*v* % BDDE-containing one after 48 h; the difference was significant. There were no significant differences between the degradation in the 5, 10, and 20 *v*/*v* % BDDE-containing matrices after 48 h, but these were all significantly different from 50G1B. There was significant difference between the 1 *v*/*v* % PEGDE-containing matrix and the 3%, and 5 *v*/*v* % PEGDE-containing ones. Thus, as the matrices were found to be stable enough, we continued to work with the 1%, 3%, and 5 *v*/*v* % BDDE (50G1B, 50G3B, and 50G5B) and PEGDE (50G1P, 50G3P, and 50G5P)-containing matrices.

### 2.4. FTIR Spectroscopy

To evaluate the FTIR spectra, we located the main characteristic peaks of gelatin. At **1029** and **1233** cm^−1^, the amide III peaks are visible; these are the C–N stretching vibrations coupled to N–H bending. The peaks at **1447** cm^−1^ and **1533** cm^−1^ are characteristic of the amide II band, which is caused by the deformation of the N-H bond [23]. At **1638** cm^−1^, the amid I region can be observed, which is associated with C=O stretching and the bending of N–H bonds with minor C–N stretching. At **3262** cm^−1^, the amide A peaks are visible, and these are due to OH stretching and N-H vibration (Figure 5) [24].

### 2.5. Free Primary Amine Content

TNBS forms trinitro benzoic groups with primary amines, which is yellow-colored and has a characteristic absorbance at 330 nm. Thus, the absorbance is directly proportional to the free primary amines. The free primary amine contents, which are the unreacted primary amine contents of the scaffolds, are shown in Figure 6. The free primary amino groups are expressed as relative absorbances compared to the absorbance of the native freeze-dried gelatin.

The free primary amino groups decreased significantly in every group that contained a crosslinker compared to native GEL even in the case of 1 *v*/*v* % BDDE and PEGDE. The 1% PEGDE crosslinked scaffolds contained significantly higher free amino groups compared to all other groups. There was no significant difference between the absorbance of the negative control and the 3 *v*/*v* % or 5 *v*/*v* % crosslinker-containing scaffolds’ absorbance.

### 2.6. Tensile Strength

Based on the results, we concluded that the optimal crosslinker was 5 *v*/*v* % BDDE; thus, we carried on with the mechanical testing of this material. The average maximum tensile load was measured and compared to the native and crosslinked GEL matrices. There were no significant differences between the different samples; thus, the crosslinked scaffolds had similar tensile strengths compared to the starting material (Figure 7).

### 2.7. Compression Test

Based on the FTIR results, as well as the swelling and degradation tests, we chose the 5 *v*/*v* % BDDE as the optimal crosslinker amount. The macroscopic properties were further evaluated using a compression test to see the mechanical difference between native gelatin and 50G5B. The results of the load deflection measurement are visible in Figure 8.

The curves represent how each material is able to withstand compression. Generally, a more rigid material has a steeper curve as the compression increases, and a softer material has a lower steepness [25]. The results were somewhat surprising: the native GEL samples were more rigid, and the crosslinked 50G5B material was softer and easier to compress; the load–compression diagram was less steep with the use of crosslinked samples.

Thus, to arrive at the optimal reaction parameters, 1 mL 5 *w*/*w* % freeze-dried gelatin matrix was used with freshly prepared BDDE/NaOH. The mixture contained either 20 µL BDDE and 300 µL 1% NaOH, or 40 µL BDDE and 600 µL 1% NaOH. In each case, the gelatin matrix was placed in the freshly prepared crosslinker mixture. Both compositions were allowed to react for 48 h at either room temperature or at 4 °C. The weight changes and swelling ratio are visible in Figure 9.

## 3. Discussion

The use of BDDE, DVS, and PEGDE as potential crosslinkers was investigated to produce a biomimetic gelatin-based scaffold that does not become soluble under physiological temperature and is suitable to be heat-sterilized without further degradation. DVS was unable to lead to a heat-stable matrix; thus, we continued with BDDE and PEGDE. According to the degradation of native GEL in alkaline solutions, we found that the presence of TRIS was more profound than the most alkaline pH in this setup, which was 12. This pH degradation screening was conducted to find the optimal pH that allows for the cleavage of the epoxy ring so the crosslinking reaction can start with the use of BDDE and PEGDE. The reaction can be enhanced with the use of catalysts [26]. However, to reduce the potential toxic materials, we chose to only use the crosslinker and simple NaOH solution, which was proven to be effective with the use of collagen between the pH values of 8.5 and 10 [27]. In our case, 5% gelatin was found to be more stable in an NaOH solution with a pH value of 12 than in a buffer of carbonate and/or TRIS, but the crosslinked material at pH = 12 further degraded during heat sterilization. Thus, although a 1% NaOH solution would completely degrade a 5% GEL solution in 4 h, we found that 1% NaOH with 1%, 3%, and 5 *v*/*v* % BDDE and PEGDE successfully crosslinked the freeze-dried 5% GEL after 48 h at room temperature, which led to a crosslinked material that could withstand heat sterilization.

Thus, we fixed the pH, the GEL content, and the reaction time and investigated the optimal amount of the crosslinkers. In order to do that, we measured the starting weight, the crosslinked weight, and the swelling ratio of the scaffolds. According to our logical explanation, degradation and crosslinking take place simultaneously; hence, the crosslinked weight is important to determine when the added crosslinker starts to increase the starting weight. Therefore, it was surprising that with the use of 1% BDDE and PEGDE, the crosslinked weight was significantly lower than the starting weight, and in the case of 3% and 5 *v*/*v* % BDDE and PEGDE, a significant weight gain was observed. However, the swelling ratios were not significantly different in either concentration: neither in the case of BDDE, nor with PEGDE. The larger weight difference in the PEGDE-crosslinked matrices compared to the BDDE matrices is probably due to the ethylene glycol chain that adds more weight to the crosslinked composition.

To test another degradation method, we used collagenase in the case of the material that was produced with 20% BDDE, and it was compared to a native 5% GEL in water and in collagenase. After we found significant differences, we carried out an enzymatic degradation screening with all the used BDDE ratios to see the effect. The use of collagenases is well known in the modeling of the in vivo degradation of biomaterials [28,29], and according to our results, the 1% and 3 *v*/*v* % BDDE-containing matrices showed the most degradation. However, the 5% BDDE-containing one was as stable as the higher BDDE-containing scaffolds, so we carried on with characterizing the 1%, 3%, and 5 *v*/*v* % BDDE-containing ones.

In the case of PEGDE, the 1 *v*/*v* % crosslinker-containing matrix degraded completely after 48 h; thus, the use of 3% and 5 *v*/*v* % is recommended.

The FTIR analysis gave us a few insights into the crosslinking reaction. In Figure 5B, we can observe that instead of the amid II band of native GEL at 1533 cm^−1^, a new peak appeared in the crosslinked matrices at 1540 cm^−1^ with the use of PEGDE. This can be explained with the partial degradation of the gelatin chains, which can lead to the disappearance of amide peaks, and the absorbances of the gelatin building blocks can appear [30]. These amino acids that build up gelatin, e.g., valine, leucine, isoleucine, and aspartic acid, have characteristic absorbance between 1502 and 1514 cm^−1^ [31]. Generally, the crosslinking via an epoxy crosslinker takes place between the carbon atom adjacent to the oxygen atom in the epoxy ring and the primary amino group [32], forming the molecules that are shown in Figure 10. In our case, these primary amines are peptides that were formed during the partial hydrolysis of collagen during the process of gelatin preparation; thus, they are presented as “R” groups.

Thus, it is probable that the absorbance of the newly formed bond appears at 1540 cm^−1^ and the amide II band is not shifted; however, there was no shift when BDDE was used. The 1447 cm^−1^ band did not shift; thus, there were probably no changes in the secondary structure of the gelatin chain [33,34,35]. It can also be observed that the intensity of the peaks at **2872** and **2934** cm^−1^, which are the C-H stretching vibrations of the CH_2_OH groups, increased compared to the native gelatin. In the amide III region, native gelatin has a peak at 1233 cm^−1^, but in the crosslinked matrices, the peak decreases, which is probably due to the changes in the secondary structure of gelatin because of the crosslinking [36]. The increased peak intensity, which appears at **1079** cm^−1^, can be identified as the C-O-C stretching vibration, which is a typical functional group in both crosslinkers. However, this indicates that the crosslinker became covalently bonded to the gelatin chain, and due to the longer chain, it was more intensive in the case of PEGDE. The crosslinker concentration difference between the 1%, 3%, and 5 *v*/*v* % BDDE did not cause the appearance or disappearance of absorbance peaks; thus, it can be concluded that the crosslinking was effective for all three concentrations. With the quantification of the free amino groups, we found that in the case of both types of the 3% and 5 *v*/*v* %, the crosslinker effectively crosslinked the free amino groups in gelatin.

However, based on the results from the collagenase enzyme-related degradation, we chose 5% BDDE to investigate the changes in the mechanical properties of native and crosslinked GEL, which is supported by the well-documented safe use of BDDE [37]. The result of the tensile strength measurements showed that the force required to tear a 2 cm-diameter freeze-dried matrix was similar to that of native gelatin. The compression test also supported this theory, as the steepness of the diagrams was also similar. The effect of crosslinker type, concentration, and other reaction parameters, like reaction time and temperature, can have a critical influence on the effectiveness of the crosslinking, which is directly connected to the crosslinked weight and the swelling ratio.

## 4. Materials and Methods

### 4.1. Hydrogel Preparation

Gelatin scaffolds were prepared at a concentration of 50 mg/mL. A total of 100 mg of gelatin (Gelita AG, Sinsheim, Germany) was weighed on an analytical balance and dissolved in 2 mL of reverse osmosis-filtered (RO) water using a Thermo-Shaker at 50 °C. The resulting solutions were lyophilized at −55 °C and 5 Pa for 24 h. Gelatin samples after freeze-drying were cut into quarters with a scalpel. BDDE (Merck, Darmstadt, Germany) and PEGDE (Merck, Germany) were mixed with a 1% *w*/*w* NaOH solution, which was used to provide an alkaline condition for the crosslinking reaction, and the mixture was pipetted onto the freeze-dried quarters. The crosslinker was used in 1% *v*/*v*, 3% *v*/*v*, 5% *v*/*v*, 10% *v*/*v*, and 20% *v*/*v* with a 150 µL NaOH solution. A scale-up step was also included, when possible; in this case, the entire 100 mg GEL-containing matrix was used and was put in the freshly mixed crosslinker/NaOH, which also contained 4 times the reagents compared to the quarters described above. The crosslinking reaction took place for 48 h at room temperature. The crosslinked gels were washed with 5 mL of RO water thrice and were freeze-dried again to reach the final form.

### 4.2. Degradation of Native Gelatin in Different pH Values

The degradation of native lyophilized ¼ matrices from 2 mL 5% *w*/*w* gelatin was measured at 7 different pH values for 168 h. The used buffers were the following: H_2_O (pH = 7), PBS (pH = 8, 0.01 M), TRIS (pH = 8, 0.01 M and pH = 9, 0.01 M), Na_2_CO_3_ and NaHCO_3_ solution (pH = 10, 0.01 M), and NaOH (pH = 11, 0.001 M and pH = 12 0.01 M). A total of 5 mL of a buffer solution was added to the native gelatin quarters, and 3 parallel measurements were taken. Gelatin leaching was then monitored using a Nanodrop UV-VIS spectrophotometer. The absorbance was measured at 205 and 230 nm.

### 4.3. Weight Differences and Swelling Ratio Measurements

For weight difference measurements, the starting freeze-dried and crosslinked freeze-dried gelatin samples were compared. Whole lyophilized gelatin matrices were weighed using an analytical balance (W_freeze-dried gel_). The gelatin quarters were then allowed to swell for 24 h and were weighed again (W_swollen gel_). The swelling ratio was calculated using the following formula:
$$Swelling\;ratio=\frac{Wswollen\;gel}{Wfreeze-dried\;gel}$$


### 4.4. Enzymatic Degradation Measurement

Two different investigations were performed. The first measurement aimed to observe and model the in vitro degradation of native and 20 *v*/*v* % BDDE-crosslinked gelatin scaffolds using 5 mL 1 mg/mL collagenase enzyme (Serva, Collagenase NB 4G) in RO water. In the second measurement, 5 mL 0.2 mg/mL collagenase was added to native and crosslinked ¼ matrices, which contained 1%, 3%, 5%, 10%, and 20% BDDE. In both cases, the samples were allowed to react on a thermostated shaker at 300 rpm and 25 °C for 48 h. The absorbances were measured with a Nanodrop One spectrophotometer at 205 and 230 nm.

### 4.5. Structural Analysis Using FTIR Spectroscopy

FTIR measurement was performed to compare the spectra of the native gelatin with the crosslinked gels. Additionally, we compared the different BDDE amounts containing gel spectra to each other in order to find out which was the most proper to stabilize gelatin. The samples were prepared as described before [19]. The measurement was performed with a Bruker Vertex 80v spectrometer. It was equipped with a high-sensitivity mercury–cadmium–telluride detector and a single-reflection diamond ATR accessory. A total of 128 scans were performed with a resolution of 2 cm^−1^ in the range of 400–4000 cm^−1^.

### 4.6. Free Primary Amine Content

The free primary amine content was measured in order to decide the effectiveness of the crosslinking. The reaction with 2,4,6-trinitrobenzenesulfonic acid (TNBS, Sigma–Aldrich, St. Louis, MO, USA) was used based on the description of Grover et al. [38] with further changes developed for our purposes, keeping in mind that we were using water-insoluble matrices. A total of 10 mg freeze-dried scaffold was used, and 1 mL carbonate buffer (pH = 10, 0.1 M) and 500 µL TNBS (10 mM) were added to the matrix. This composition was allowed to react at 30 °C for 30 min, and then 1 mL SDS (10 *m*/*m*%) and 500 µL HCl (1 M) were added to the mixture. The absorbance was measured at 335 nm with the use of a UV–Vis spectrophotometer (Biotek Powerwave XS, Winooski, VT, USA).

### 4.7. Compression Test

To test the compressive properties of the scaffolds, a compression test was performed using Instron 5566 universal testing machine (Instron, Norwood, MA, USA). A capacity of 500 N load cell was used, and the speed of the crosshead was set to 1 mm/min. Native and crosslinked 5% BDDE-containing gelatin scaffolds were used, and 3 parallel measurements were taken.

### 4.8. Tensile Strength

Tensile tests were performed on an Instron 5566 universal testing machine (Instron, Norwood, MA, USA) with a capacity of 500 N load cell. The gauge length was set to 12 mm, and a crosshead speed of 2 mm/min was used. Three parallel measurements were carried out on each sample.

### 4.9. Statistical Analysis

One-way and two-way analysis of variance (ANOVA) were performed with a Tukey’s post hoc test to compare differences between the groups. The significance level was *p* > 0.05, where * means that *p* is between 0.01 and 0.05, ** means that *p* is between 0.01 and 0.001, and *** means that *p* is lower than 0.001. Prism 7 software (Irvine, CA, USA) was used for statistical analysis. Data are presented as mean ± SEM.

## 5. Conclusions

The present investigation focused on the optimal reaction parameters in the construction of gelatin-based scaffolds. Freeze-dried gelatin matrices were found to be the optimal starting materials, which were placed in the crosslinker mixture, and the crosslinking was allowed to take place for 48 h at 4 °C. Based on our experiments, the most efficient crosslinking was under alkaline conditions in 1% NaOH. The most promising reagents were the matrices, which were prepared with the use of 1%, 3%, and 5 *v*/*v* % BDDE and PEGDE. According to the enzymatic degradation measurements, 5 *v*/*v* % BDDE matrices were the best, with 48 h of crosslinking at 4 °C. To prove the formation of covalent bonds, FTIR was used, which confirmed the presence of C-O-C groups. The mechanical strength of the materials was found to be similar to that of the starting gelatin matrix. The scaffolds were water-insoluble, resistant to collagenase enzyme, and able to withstand heat sterilization. Thus, a further aim is the application of the crosslinked matrix in vitro to see if hMSCs could adhere to and proliferate on the matrix. Ultimately, the matrix is intended to be used for medicinal purposes as a soft tissue implant that can be an important tool in regenerative medicine.

## Figures and Tables

**Figure 1 ijms-25-04336-f001:**
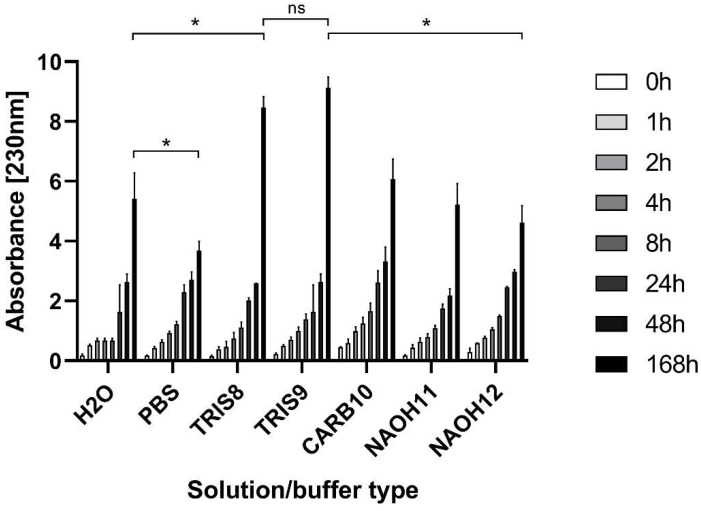
The degradation of native freeze-dried gelatin; the significance level was *p* < 0.05. All data are presented as mean ± standard error of the mean (*n* = 3). ns: not significant, * is explained in the statistical analysis chapter.

**Figure 2 ijms-25-04336-f002:**
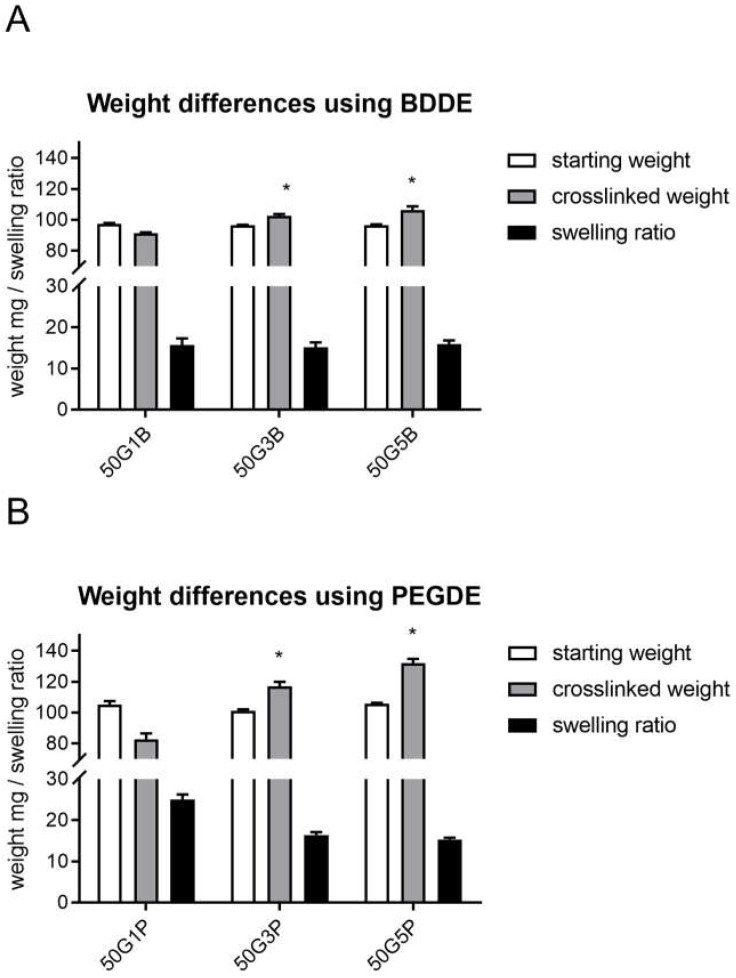
Weights of the freeze-dried starting gelatin and the freeze-dried crosslinked gelatin scaffolds, including the swelling ratio after 48 h of crosslinking, with the use of 1%, 3%, and 5 *v*/*v* % BDDE (**A**) and PEGDE (**B**) in 600 µL of 1% NaOH. The significance level was *p* < 0.05. All data are presented as mean ± standard error of the mean (*n* = 3). * is explained in the statistical analysis chapter.

**Figure 3 ijms-25-04336-f003:**
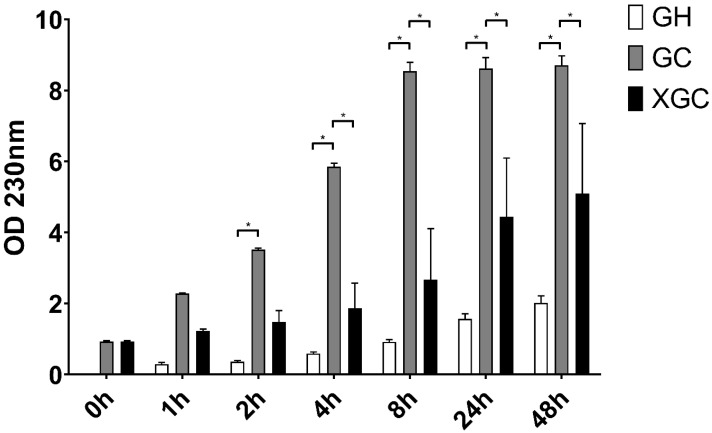
Comparison of the degradation of native GEL in H_2_O (GH), native GEL in 1 mg/mL collagenase (GC), and 20 *v*/*v* % BDDE crosslinked GEL in 1 mg/mL collagenase (XGC) at 230 nm. The significance level was *p* < 0.05. All data are presented as mean ± standard error of the mean (*n* = 3). * is explained in the statistical analysis chapter.

**Figure 4 ijms-25-04336-f004:**
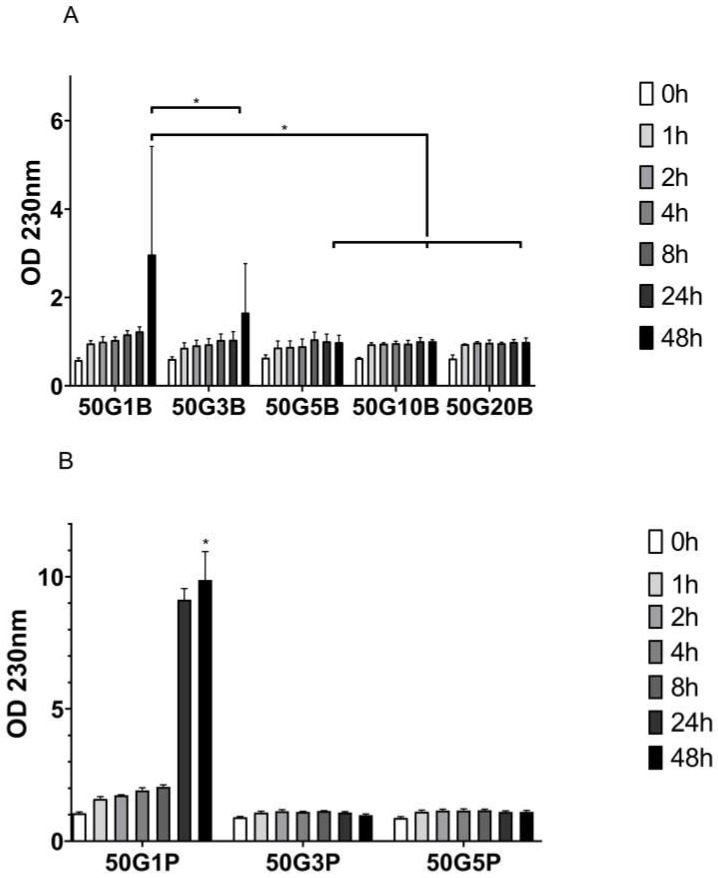
Comparison of the degradation of crosslinked gelatin with 1%, 3%, 5%, 10%, and 20 *v*/*v* % BDDE (**A**) and 1%, 3%, and 5 *v*/*v* % PEGDE (**B**) in 0.2 mg/mL collagenase at 230 nm. The significance level was *p* < 0.05. All data are presented as mean ± standard error of the mean (*n* = 3). * is explained in the statistical analysis chapter.

**Figure 5 ijms-25-04336-f005:**
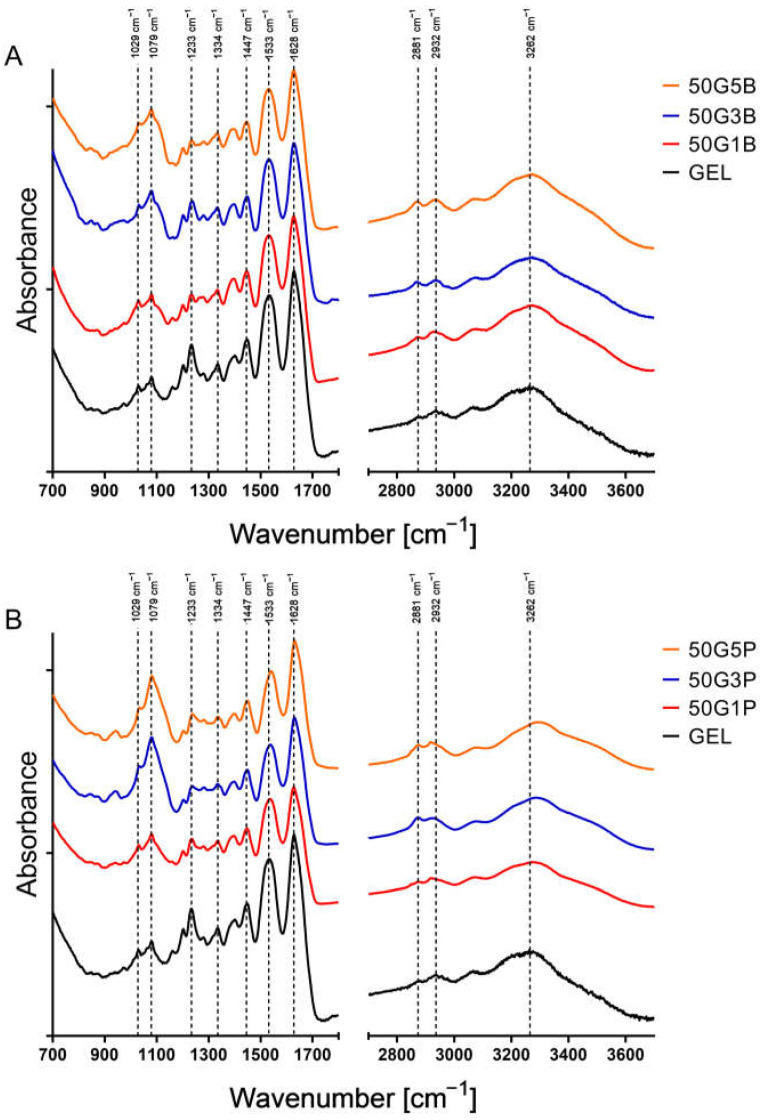
FTIR spectra of native and crosslinked gelatin. GEL is visible in the bottom of the figure, and from bottom to top, the 1%, 3%, and 5 *v*/*v* % BDDE (**A**) and 1%, 3%, and 5 *v*/*v* % PEGDE (**B**) crosslinker-containing matrices are visible.

**Figure 6 ijms-25-04336-f006:**
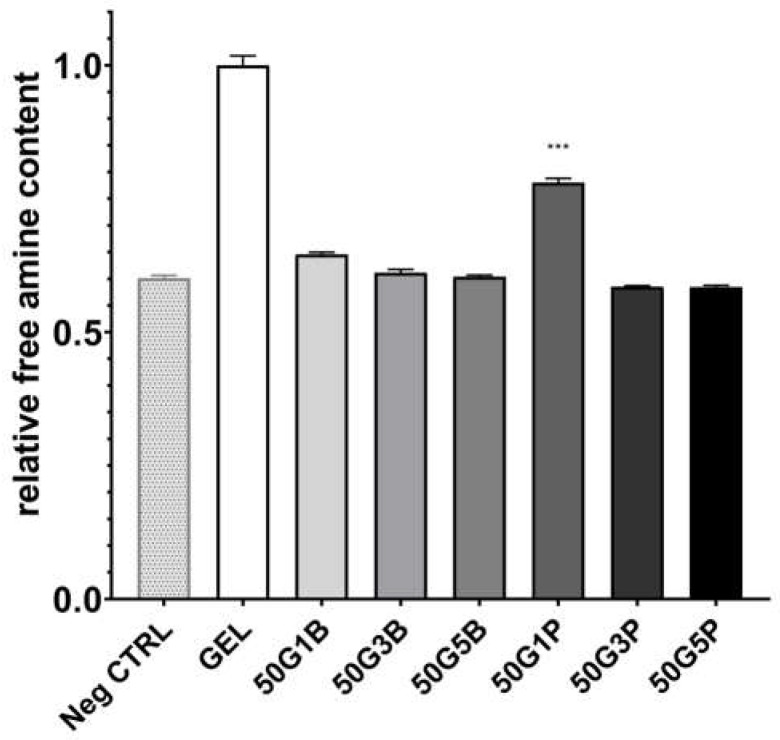
Relative absorbance of native and crosslinked gelatin matrices at 335 nm. The absorbances are expressed as relative absorbances compared to native freeze-dried gelatin (*n* = 3). *** is explained in the statistical analysis chapter.

**Figure 7 ijms-25-04336-f007:**
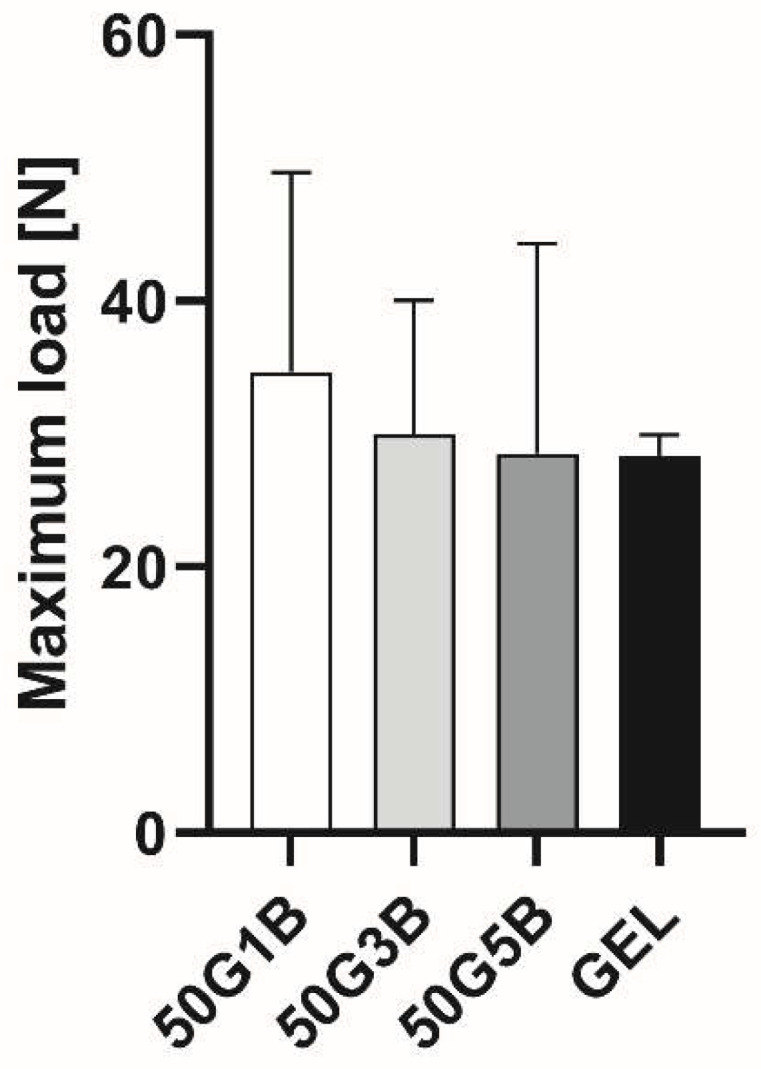
Average maximum tensile load of native GEL scaffolds and 1%, 3%, and 5 *v*/*v* % BDDE-crosslinked GEL. Both native GEL samples and crosslinked samples were tested in triplicate.

**Figure 8 ijms-25-04336-f008:**
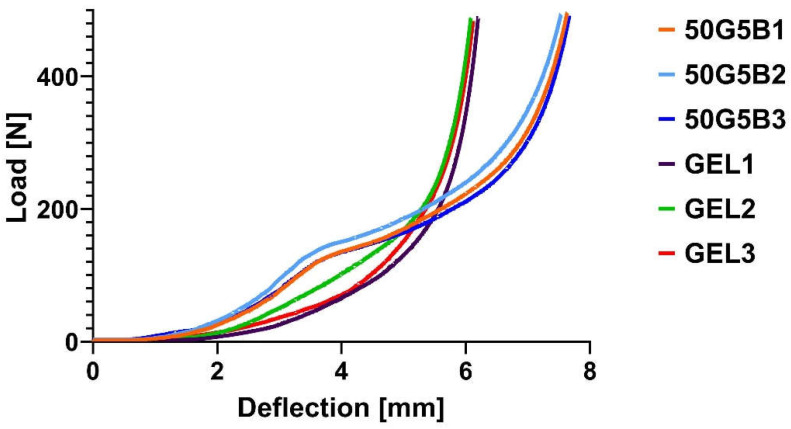
Load–deflection curves of native GEL scaffolds and 5 *v*/*v* % BDDE-crosslinked GEL (50G5B). Both native GEL samples and crosslinked samples were tested in triplicate.

**Figure 9 ijms-25-04336-f009:**
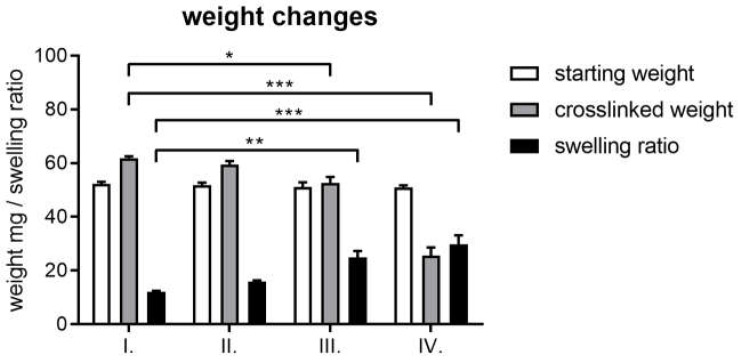
Comparison of crosslinked weight differences and swelling ratios using 20 µL BDDE in 300 µL 1% NaOH at 4 °C for 48 h (I.), 40 µL BDDE in 600 µL 1% NaOH at 4 °C for 48 h (II.), 20 µL BDDE in 300 µL 1% NaOH at RT for 48 h (III.), and 40 µL BDDE in 600 µL 1% NaOH at RT for 48 h (IV.). *,**,*** are explained in the statistical analysis chapter.

**Figure 10 ijms-25-04336-f010:**
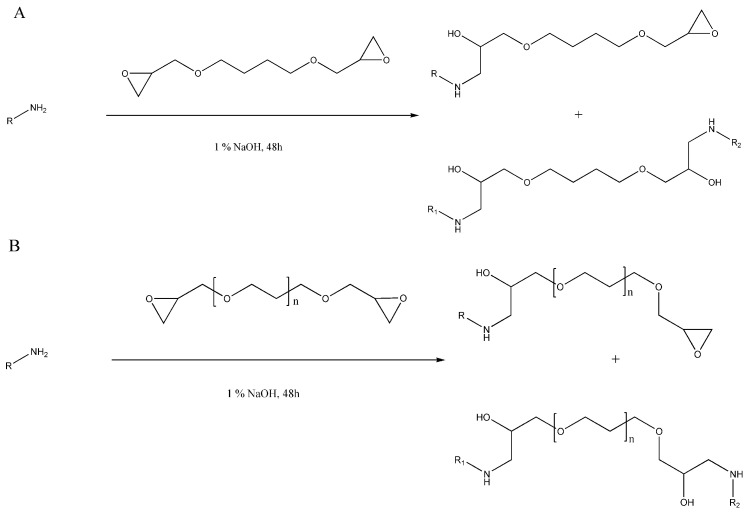
General reaction between a primary amino group-containing molecule and BDDE (**A**) or PEGDE (**B**) with the expected products.

## Data Availability

The data presented in this study are available on request from the corresponding author.

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
