# Peer review of "Water-Insoluble, Thermostable, Crosslinked Gelatin Matrix for Soft Tissue Implant Development"

_ijms, 2024, doi:10.3390/ijms25084336_

Round 1
Reviewer 1 Report
Comments and Suggestions for Authors
Please see the attached document.

Comments on the Quality of English LanguagePlease see the attached document.
Author Response
We are thankful for the grammar corrections of the reviewer, the corrections were put in the manuscript.
- Page 2: Do proliferate and grow mean the same thing here?
Yes, grow and proliferate mean the same thing here, thus the sentence was corrected.
- Page 3: Should add this data as supplementary materials
The experiments that helped us in the selection of 48 hours do not contain important information that would need a supplementary file. Thus, it was simply added to the text that after 24 hours the crosslinked weight was lower than the starting weight.
- Page 4: Data should be shown as supplementary materials
This experiment did not seem to be significant enough to be added as a separate supplementary file, thus, simply the statement was left in text that indicates that the swelling ratio didn’t increase above 5 % V/V crosslinker.
Reviewer 2 Report
Comments and Suggestions for Authors
1. General observation on the manuscript: the sentences are too long, making it difficult to go through the text, and from the point of view of English, I think the text should be revised from this point of view.
2. Lines 94-98 should be corrected. The objective of the work must be explicitly defined because as it is written, it is not clearly understood.
3. The introduction generally makes several references to the studies of the authors of this manuscript and should include other studies that refer to the thermoversability of gelatin. It must also be demonstrated that other studies have not addressed such a subject.
4. I recommend the authors reduce the number of self-citations in the manuscript, for example, the bibliography contains quite a lot of self-citations (references 9, 21, 20, 6, 7, 8, 2, 3, 4).
5. Lines 123-124: Why do the authors consider that the reaction time of 48 h is optimal? Some additional explanations should be included.
6. What do you mean when you use the expression "crosslinked weights"?
7. What do you mean when you use the expression "crosslinked weights"? Please review the text from lines 150-154.
8. Figure 3: it is not clear what the legend represents. How do you explain the relatively large errors for the samples analyzed at 8, 24, and 48 hours?
9. Figure 4: please indicate A or B as specified in the caption, likewise for Figure 5.
10. What is the reason for the decrease of amino groups, after using a crosslinking agent? What is the chemical reaction taking place? Please elaborate. It is not enough just to observe a phenomenon, it must be explained.
11. Lines 246-250, you must include a bibliographical reference to support your claim.
12. Row 400: please correct "1 ml carbonate puffer"
13. The 440-443 should be corrected. Please shorten the sentence.
14. The authors claim that they choose low-toxicity crosslinkers.
15. The conclusion section is written more like a story not in a scientific manner. Review this section and present your most important observations following the objectives of the paper, as well as the results obtained.
For example: "The chemical modification was assessed with FTIR and the compression and tensile strength was also tested" - this already was specified in the method section.
Comments on the Quality of English Language
The sentences are long and very hard to follow. In some cases, words like: "are the best" are used.
Author Response
We are grateful for the comments of theReviewer and the opportunity to revise the paper and to improve our work to reach the required scientific level of our work. We have considered the notes and suggestions of the Reviewer, please find the answers below.
Reviewer 2
- General observation on the manuscript: the sentences are too long, making it difficult to go through the text, and from the point of view of English, I think the text should be revised from this point of view.
The text was revised and corrected according to the suggestion of the reviewer.
- Lines 94-98 should be corrected. The objective of the work must be explicitly defined because as it is written, it is not clearly understood.
The sentence was modified as requested.
- The introduction generally makes several references to the studies of the authors of this manuscript and should include other studies that refer to the thermoversability of gelatin. It must also be demonstrated that other studies have not addressed such a subject.
Two further articles were added that describe crosslinking and thermo-reversibility.
- I recommend the authors reduce the number of self-citations in the manuscript, for example, the bibliography contains quite a lot of self-citations (references 9, 21, 20, 6, 7, 8, 2, 3, 4).
The least relevant self-citations were removed, and a duplicate was also found and removed.
- Lines 123-124: Why do the authors consider that the reaction time of 48 h is optimal? Some additional explanations should be included.
The crosslinked weight was lower than the starting weight after 24 hours, but after 48 hours, the crosslinked weight exceeded the starting weight, thus it was found to be optimal. The text was modified with a brief info on this.
- What do you mean when you use the expression "crosslinked weights"?
The weight of the crosslinked gelatin scaffolds, the explanation was added below Figure 2.
- What do you mean when you use the expression "crosslinked weights"? Please review the text from lines 150-154.
The weight of the crosslinked gelatin scaffolds, the explanation was added below Figure 2.
- Figure 3: it is not clear what the legend represents. How do you explain the relatively large errors for the samples analyzed at 8, 24, and 48 hours?
This experiment was conducted with a relatively large amount of collagenase enzyme (1 mg/ml) that led to a heterogenous degradation, thus in the next steps, reduced amount of collagenase was used (0.2 mg/ml).
- Figure 4: please indicate A or B as specified in the caption, likewise for Figure 5.
The “A” and “B” parts were specified in the images as requested by the reviewer in Figure 2, Figure 4, and Figure 5.
- What is the reason for the decrease of amino groups, after using a crosslinking agent? What is the chemical reaction taking place? Please elaborate. It is not enough just to observe a phenomenon, it must be explained.
The epoxy crosslinkers can form covalent bonds between the free primary amino groups, thus, with the reduction of the free primary amino groups we can prove that the crosslinking did occur. The text was modified according to this.
- Lines 246-250, you must include a bibliographical reference to support your claim.
We have added a relevant scientific article entitled “Load-compression behavior of brittle foams” as a reference according to the suggestion of the reviewer.
- Row 400: please correct "1 ml carbonate puffer"
The mistake was corrected according to the reviewer.
- The 440-443 should be corrected. Please shorten the sentence.
Two separate sentences were created instead of one long, as the reviewer requested.
- The authors claim that they choose low-toxicity crosslinkers.
Yes, BDDE, and generally the epoxy crosslinkers are one of the most commonly used crosslinkers, which have already been used to crosslink hyaluronic acid for over 20 years.
- The conclusion section is written more like a story not in a scientific manner. Review this section and present your most important observations following the objectives of the paper, as well as the results obtained.
The conclusion was modified, in a more objective manner, the main findings were collected as it was suggested by the reviewer.
For example: "The chemical modification was assessed with FTIR and the compression and tensile strength was also tested" - this already was specified in the method section.
Reviewer 3 Report
Comments and Suggestions for Authors
Reviewer’s comments:
The manuscript entitled ‘Water insoluble, thermo-stable, crosslinked gelatin matrix for scaffold development’ has been peer-reviewed. In the present work, the authors have prepared a crosslinked gelatin matrix to improve the scaffold’s properties. The manuscript should be rejected due to lack of novelty, incomprehensible grammatical sentences and technical words, insufficient data, and unimpressive presentation. The following comments can be useful to improve the manuscript.
Rejection
Several incomprehensible statements from the title to conclusion have been found in the manuscript.
1) In the title and abstract, the author should specify the reason why the scaffold was developed. Are there any biomedical applications?
2) Abstract.
a) In the present study the material science background of crosslinked gelatin was investigated.
The background to the current work should be provided. But the author has mentioned that the background of crosslinked gelatin was investigated.
b) The optimal reaction parameters were assessed for the production of a water insoluble crosslinked gelatin matrix that is suitable for heat sterilization.
Whether is it ‘suitable for heat sterilization’ or ‘stable against heat’?
c) The matrices were found to withstand enzymatic degradation, and in the case of butanediol diglycidyl ether (BDDE) and poly(ethylene glycol) diglycidyl ether (PEGDE) also heat sterilization.
Whether the authors would like to reveal that all the prepared scaffolds are stable against heat and enzymatic degradation?
d) We also found that in the case of 1 V/V% crosslinker, the crosslinked weight was lower than the starting weight, thus probably degradation and crosslinking happen simultaneously.
The author can use ‘initial weight’ instead of ‘starting weight’.
What is the name of crosslinker?
Whether ‘the crosslinked weight’ means ‘the weight of crosslinked gelatin’?
Whether enzymatic degradation favors crosslinking as both processes occur simultaneously?
e) In the case of 3, and 5 V/V % BDDE and PEGDE the crosslinked weight and the swelling ratio was optimal, the crosslinking was confirmed with the use of FTIR, and the absence of free primer amino groups also proved the effectiveness of the crosslinking, even in the case of 1 V/V % crosslinker.
Is it the ‘primary’ amino groups present in gelatin?
f) We found that the stress strain and compression characteristics of the 5 V/V % BDDE crosslinked matrix are comparable to the native gelatin (GEL), thus based on the material science measurements, the crosslinked matrices may be suitable candidates for scaffold development.
Whether the author would like to reveal that the crosslinked gelatin matrix exceeded native gelatin in the aspects of stress, strain, and compression properties?
Even the uncrosslinked gelatin can be utilized as a scaffold. The author can mention here that the crosslinked gelatin matrices exhibit superior physicochemical and mechanical properties than native gelatin, and hence can be suggested as a promising scaffold in tissue regeneration.
3) The fastest decomposition was observed in the case of TRIS8 and TRIS9 solutions, the degradation in these buffers were significantly higher compared to all other groups, however there was no significant difference between the degradation in the two TRIS buffer solutions.
‘Decomposition’ differs from ‘degradation’. The author should use a suitable technical word to describe it.
4) Figure 1. The degradation of native freeze-dried gelatin.
Is it not a crosslinked gelatin?
5) The author should explain how TNBS reagent is helpful to determine free amino acid groups in gelatin, that can be measured by UV-vis absorbance at 335 nm?
6) The authors should provide a section of ‘Materials’ to specify the source of availability of chemicals.
7) If the scaffold can be prepared for tissue engineering and regenerative medicine, then the authors should report cell line studies.
8) The author should specify the novelty of the present work in the introduction comparing it with previous reported literatures.
9) Figure 3 is not properly placed, and some contents are hidden.
10) Figure 10. The name of each substance can be indicated for better understanding.
Comments on the Quality of English Language
Incomprehensive English sentences throughout the manuscript.
Author Response
We are grateful for the Reviewer’s comments and the opportunity to revise the paper and to improve our work to reach the required scientific level of our work. We have considered the notes and suggestions of the Reviewer, please find the answers below.
Several incomprehensible statements from the title to conclusion have been found in the manuscript.
1) In the title and abstract, the author should specify the reason why the scaffold was developed. Are there any biomedical applications?
The title and abstract were updated with the planned intended use as requested by the reviewer. The material is planned to be regulated as a Class III medical device.
2) Abstract.
- a) In the present study the material science background of crosslinked gelatin was investigated.
The background to the current work should be provided. But the author has mentioned that the background of crosslinked gelatin was investigated.
The type of crosslinking that we used was not tested on gelatin yet, only the crosslinking of collagen was done before. A brief info was added to the text according to the suggestion of the reviewer.
- b) The optimal reaction parameters were assessed for the production of a water insoluble crosslinked gelatin matrix that is suitable for heat sterilization.
Whether is it ‘suitable for heat sterilization’ or ‘stable against heat’?
From the viewpoint of medical device development, “suitable for sterilization” is the better one, as the product needs to be sterilized for the end user.
- c) The matrices were found to withstand enzymatic degradation, and in the case of butanediol diglycidyl ether (BDDE) and poly(ethylene glycol) diglycidyl ether (PEGDE) also heat sterilization.
Whether the authors would like to reveal that all the prepared scaffolds are stable against heat and enzymatic degradation?
The scaffolds that were selected for heat sterilization were the ones that were able to withstand the enzymatic degradation, so heat sterilization was based on the results of the collagenase experiment.
- d) We also found that in the case of 1 V/V% crosslinker, the crosslinked weight was lower than the starting weight, thus probably degradation and crosslinking happen simultaneously.
The author can use ‘initial weight’ instead of ‘starting weight’.
We decided to call the weight of the gelatin that was originally measured before the melting and freeze-drying procedure “initial weight” thus, we named the weight before the crosslinking the “starting weight”.
What is the name of crosslinker?
The two epoxy crosslinkers were: butanediol diglycidyl ether (BDDE) and poly(ethylene glycol) diglycidyl ether (PEGDE).
Whether ‘the crosslinked weight’ means ‘the weight of crosslinked gelatin’?
Yes, the “crosslinked weight” means the weight of the crosslinked, freeze-dried gelatin scaffolds.
Whether enzymatic degradation favors crosslinking as both processes occur simultaneously?
Enzymatic degradation acts on the peptide bonds, thus probably the enzyme cannot reach the peptide bonds due to the newly formed covalent bonds, which do not allow the enzyme to effectively reach the peptide bonds sterically.
- e) In the case of 3, and 5 V/V % BDDE and PEGDE the crosslinked weight and the swelling ratio was optimal, the crosslinking was confirmed with the use of FTIR, and the absence of free primer amino groups also proved the effectiveness of the crosslinking, even in the case of 1 V/V % crosslinker.
Is it the ‘primary’ amino groups present in gelatin?
Yes, the reviewer is right, it is “primary” amine, and the text was corrected in the manuscript.
- f) We found that the stress strain and compression characteristics of the 5 V/V % BDDE crosslinked matrix are comparable to the native gelatin (GEL), thus based on the material science measurements, the crosslinked matrices may be suitable candidates for scaffold development.
Whether the author would like to reveal that the crosslinked gelatin matrix exceeded native gelatin in the aspects of stress, strain, and compression properties?
Even the uncrosslinked gelatin can be utilized as a scaffold. The author can mention here that the crosslinked gelatin matrices exhibit superior physicochemical and mechanical properties than native gelatin, and hence can be suggested as a promising scaffold in tissue regeneration.
Unfortunately, the stress-strain, and compression were similar to the starting freeze-dried gelatin matrix, so they did not exceed the starting gelatin. The advance of the crosslinked matrix is the water insolubility, reduced enzymatic degradation, and the ability to withstand sterilization, and physiological temperature, without melting.
3) The fastest decomposition was observed in the case of TRIS8 and TRIS9 solutions, the degradation in these buffers were significantly higher compared to all other groups, however there was no significant difference between the degradation in the two TRIS buffer solutions.
‘Decomposition’ differs from ‘degradation’. The author should use a suitable technical word to describe it.
The reviewer is right, degradation is the correct word, and it was corrected in the text as well.
4) Figure 1. The degradation of native freeze-dried gelatin.
Is it not a crosslinked gelatin?
No, this is the native freeze-dried gelatin without crosslinking.
5) The author should explain how TNBS reagent is helpful to determine free amino acid groups in gelatin, that can be measured by UV-vis absorbance at 335 nm?
A brief description of the TNBS reaction was added to the Results section to the “Free primary amine content” paragraph.
6) The authors should provide a section of ‘Materials’ to specify the source of availability of chemicals.
The reviewer is right, the supplier of the main components was added to the Materials and Methods section.
7) If the scaffold can be prepared for tissue engineering and regenerative medicine, then the authors should report cell line studies.
The present manuscript focuses mainly on the material science and selection of the scaffold that is potentially suitable to be used in vitro and later in vivo. The in vitro studies are currently in a pre-liminary state.
8) The author should specify the novelty of the present work in the introduction comparing it with previous reported literatures.
The manuscript was modified with a brief novelty description in the introduction section according to the suggestion of the reviewer.
9) Figure 3 is not properly placed, and some contents are hidden.
The image was resized, according to the suggestion of the reviewer.
10) Figure 10. The name of each substance can be indicated for better understanding.
The substances cannot be exactly defined, as the primary amines are peptides that were formed during the partial hydrolysis of collagen during the process of gelatin preparation, and probably during the crosslinking additional degradation occurs, thus they are presented as “R” groups. This information was put in the text.
Round 2
Reviewer 3 Report
Comments and Suggestions for Authors
The authors have modified the manuscript with appropriate changes following the reviewer's comments. The manuscript can be accepted after addressing the following minor inquiry.
The abstract should have background/aim, characterization, results and discussion, and conclusion in order. The authors should ensure that the abstract contains all the content.
Comments on the Quality of English Language
Minor Editing of English may be required.
Author Response
We would like to thank the suggestions of the Reviewer, the abstract was corrected and rearranged as can be seen below:
The manuscript was updated with this text, and the grammar was checked and corrected again.
Background/Aim:
In the present study, the material science background of crosslinked gelatin (GEL) was investigated. The aim was to assess the optimal reaction parameters for the production of a water insoluble crosslinked gelatin matrix suitable for heat sterilization.
Characterization:
Matrices were subjected to enzymatic degradation assessments, and their ability to withstand heat sterilization was evaluated. The impact of different crosslinkers on matrix properties was analyzed.
Results and Discussion:
It was found that matrices crosslinked with butanediol diglycidyl ether (BDDE) and poly(ethylene glycol) diglycidyl ether (PEGDE) were resistant to enzymatic degradation and heat sterilization. Additionally, at 1 V/V% crosslinker concentration, the crosslinked weight was lower than the starting weight, suggesting simultaneous degradation and crosslinking. The crosslinked weight and swelling ratio were optimal in the case of the matrices that were crosslinked with 3% and 5% V/V BDDE and PEGDE. FTIR analysis confirmed crosslinking, and the reduction of free primary amino groups indicated effective crosslinking even at 1% V/V crosslinker concentration. Moreover, stress-strain and compression characteristics of the 5% V/V BDDE crosslinked matrix were comparable to native gelatin.
Conclusion:
Based on material science measurements, the crosslinked matrices may be promising candidates for scaffold development, including properties such as resistance to enzymatic degradation and heat sterilization.
We hope that the reviewer will be satisfied with our work.
